# Economic Growth and Cardiorespiratory Fitness of Children and Adolescents in Urban Areas: A Panel Data Analysis of 27 Provinces in China, 1985–2014

**DOI:** 10.3390/ijerph16193772

**Published:** 2019-10-08

**Authors:** Xiaomei Gan, Xu Wen, Yijuan Lu, Kehong Yu

**Affiliations:** 1Department of Sport Science, College of Education, Zhejiang University, Hangzhou 310000, China; 11603020@zju.edu.cn (X.G.); wenxu@zju.edu.cn (X.W.); 11703023@zju.edu.cn (Y.L.); 2Center for Sports Modernization and Development, Zhejiang University, Hangzhou 310000, China

**Keywords:** children and adolescents, economic growth, cardiorespiratory fitness, relationship

## Abstract

With rapid economic development in China, cardiorespiratory fitness (CRF) of children and adolescents is on a decline. However, this appears to have slowed down, reaching stagnation in certain areas. However, it is unclear if the change in CRF is related to economic growth and development or not. This study describes trends in CRF of Chinese children and adolescents, and empirically tests the relationships between China’s macro-economic developments and cardiorespiratory fitness of children and adolescents over the past 30 years using provincial panel data collected from one million samples. We used per capita disposable income as the economic indicator. CRF was assessed by using running tests: 50 m × 8 for boys and girls (7–12 years), 1000 m for boys (13–22 years), and 800 m for girls (13–22 years). The results show that economic growth has a U-shaped relationship with CRF of children and adolescents (both boys and girls). It appears that as incomes increased, CRF of urban male and female students in China gradually decreased to its lowest point, after which it showed an upward trend. From a horizontal perspective, it can be inferred that for low-developed provinces, increases in incomes cause a decrease in CRF levels. In contrast, for highly developed provinces, as incomes increase, CRF levels increase. This study provides the first empirical evidence of the relationship between macro-economy and CRF of youth, based on provincial panel data. The results presented here can be used to formulate health policies targeting the cardiorespiratory fitness of children and adolescents from middle-income provinces in China. This study also provides a reference for developing countries.

## 1. Introduction

Low cardiorespiratory fitness (CRF) is a strong and independent predictor of cardiovascular disease [1], cancer [2], and diabetes [3]. A previous study reported that global physical activity is on a decline [4]. In recent decades, the US, Canada, France, Australia, Italy, South Korea, Finland, and Norway, among other countries, have witnessed a significant decrease in cardiorespiratory fitness of children and adolescents [5,6,7,8]. Sedentary lifestyles and lack of physical activity are thought to be the main causes for this decline [9,10]. Moreover, social-economic environments, technology development, urbanization, and urban development influence CRF more than other public health factors [11,12].

A cross-sectional study on cardiorespiratory fitness in children and adolescents from several countries concluded that there is a strong negative correlation between national income inequality (GiNi index) and CRF of children and adolescents [13,14]. In low- and middle-income countries, high urbanization and incomes are risk factors for chronic cardiovascular related-diseases [15]. However, the growth of national economies has altered the relationship between urbanization and individual physical activity. For example, from 1991 to 2009, the level of physical activity of individuals living in highly urbanized areas in China was lower than that of those living in less-urbanized areas. However, research has revealed that this difference diminishes with rise in incomes in less-urbanized areas [15]; intensive construction of residential buildings [16], sports facilities, and transportation facilities [17] also contributes to this change.

Previous studies have explored the impact of socio-economic development on physical activity and cardiopulmonary function, primarily using cross-sectional data. Due to the lack of time series data, the impact of various CRF factors has not been tested. Moreover, few studies have investigated CRF from the perspective of socio-economic development. Although China’s economy has grown rapidly over the past 30 years, the country is facing several challenges, especially in the social security, and medical and health sectors, which require new reforms and policies to improve the health of the population. Based on panel data from 27 provinces in China collected over the past 30 years, this study analyzed the CRF of one million children and adolescents aged 7 to 22 years. The authors aimed to describe the trends in CRF of children and adolescents, evaluate the impact of rapid development of social economy on CRF using provincial panel data, and explore the relationship between economic development and CRF of children and adolescents in a developing country. To the best of our knowledge, this perspective has not been reported before. Our results provide empirical evidence of the association between economy and children’s CRF, and explore the impact of economy on students’ CRF. These findings provide a scientific basis for policy formulation and effective intervention, and also serve as a reference for developing countries.

## 2. Method

### 2.1. Data

#### 2.1.1. CRF Data

CRF data were derived from the Chinese National Surveys on Students’ Constitution and Health (CNSSCH) conducted in 1985, 1991, 1995, 2000, 2005, 2010, and 2014 [18,19,20,21,22,23,24]. These series of surveys were conducted by the Ministries of Education, Health, Science and Technology; the State Ethnic Affairs Commission; and the State Sports General Administration of the People’s Republic of China. This study only included students of Han ethnicity, which constitutes 92% of the total Chinese population. The respondents were from 27 of the 31 provinces in China, excluding Hainan and Chongqing, both of which were founded after 1985. Qinghai and Tibet autonomous regions were also excluded because Qinghai was not included in the 1995 survey and Tibet autonomous region was not covered in nearly all surveys. Each study enrolled an equal number of students from each province. The participants were aged 7–22 years (primary to college level), and were selected from the same areas in each province from 1985 to 2014. The students were selected using the stratified cluster sampling method from certain classes, and clusters were randomly selected from each grade in the selected schools. Cardiorespiratory fitness was assessed using running tests: 50 m × 8 for girls and boys aged 7–12 years, 1000 m for boys aged 13–22 years, and 800 m for girls aged 13–22 years. Table 1 shows the sample sizes at each examination period. The research protocol was reviewed and approved by the Ethics Committee of College of Education at Zhejiang University.

#### 2.1.2. Socioeconomic Data

The data of per capita disposable income (PCDI) were derived from the respective statistical yearbooks of China’s provinces (1985, 1991, 1995, 2000, 2005, 2010, and 2014) and the statistical database of the Chinese economic network [25].

### 2.2. Variables

To overcome heteroscedasticity and enhance the stationarity of data, the logarithm of all variables was used [26]. The dependent variables were L50 × 8, L1000 and L800, which was the log of the student’s running test time of 50 m × 8, 1000 m, and 800 m. As the core independent variable, PCDI served as the economic indicator. The control variables were urbanization rate (URBAN) and consumer price index (CPI). Considering that urbanization correlates with physical activity [14,15], the rate of urbanization was used to determine its impact on CRF. In addition, over-nutrition was associated with a decreased cardiorespiratory endurance level [27]; thus, CPI was utilized to assess the indirect impact of nutrition on CRF. Table 2 illustrates the variables used in this study.

### 2.3. Estimation Approach

Panel econometric models were employed to estimate the relationship between students’ CRF status and PCDI. To reduce endogeneity and increase the scientific soundness of the model, several control variables were introduced. The models used are as follows:

Equation (1) is used to estimate the relationship between economic development and the students’ CRF. Empirical analysis of the static model was performed using Stata 12.0 software (StataCorp, College Station, Texas, USA): LY = α + β_1_ × LPCDI + β_2_ × LURBAN + β_3_ × LCPI + t + ε(1)
where LY is log of the CRF status of students at each age in each province and survey, LPCDI is the log of economic level of each province and survey, LURBAN stands for log of urbanization rate of each province and survey, and LCPI represents the log of the consumer price index of each province and survey. LURBAN and LCPI are control variables that are expected to be related to the CRF status, α represents the fixed effects of a province, t is the time-specific effects, and ε is the error term. These equations employ the fixed effect regression model, which controls for time-invariant characteristics, such as climatic conditions and unmeasured cultural factors, and any time-varying differences common to all provinces.

Equation (2) is the quadratic model used to investigate the marginal effect of economic growth on the students’ CRF, that is, to verify whether a non-linear relationship exists between the two:LY = α + β_1_ × LPCDI + β_2_ × LPCDI^2^ + β_3_ × LURBAN + β_4_ × LCPI + t + ε(2)

## 3. Results

### 3.1. Secular Trends in CRF from 1985 to 2014

The secular trends of mean CRF test time for boys and girls from 1985 to 2014 are shown in Table 3. The surveys focused on 7, 13, 16, and 19-year-old students as the specific ages for first-grade, junior high school, senior high school, and university levels, respectively. The CRF of boys at almost all ages showed a continuous decreasing trend from 1985 to 2014, except for the seven-year-old boys, whose CRF rebounded from 2005 to 2014. From 1985 to 2005, the CRF of girls at all ages showed a decreasing trend, but from 2005 to 2014, the CRF of girls at almost all ages rebounded, except for those aged 19 in college. The biggest decline in CRF for both boys and girls occurred at age 13. The time taken to run 1000 m and 800 m increased by 37.1 s and 30.7 s, respectively, during the 30-year period.

### 3.2. Empirical Analysis

The fixed effects model regression was applied to all models. The fixed effects model was chosen because time-invariant variables were not included in the study. It offers the advantage of not assuming the relationship between the error term and the explanatory variable [26]. On the basis of the natural logarithm of each variable, robust command was used to correct the standard error with white heteroscedasticity to ensure robust results. The subsequent regressions eliminated the outliers.

#### 3.2.1. Linear Analysis

Columns (linear) in Table 4 and Table 5 represent the estimated results for Equation (1), which show the performance of linear specification between economic growth in China and CRF of children and adolescents after controlling for other variables, such as urbanization rate and CPI. In Table 4, the coefficients of LPCDI for the CRF of boys and girls (7–12-year-old) are −0.006 and −0.001, small and not significant (*p* > 0.1), indicating that there is no linear relationship between PCDI and CRF of children living in urban areas. Table 5 shows that the model yielded a significantly positive coefficient of LPCDI for boys (13–22-year-old): −0.027. Given the log-log specification, the coefficients represent the elasticity of the mean running time with respect to PCDI. A 1% increase in per capita PCDI is accompanied by a decrease in mean running time by 0.027% for boys (aged 13–22 years). This finding implies that PCDI has a positive impact on the cardiopulmonary fitness of boys (aged 13–22 years), contrary to the descriptive results in Table 3, and theoretical reasoning. This seems to be related to the fact that part of the decrease in CRF is attributed to a positive time trend for any nationwide factor (e.g., national economic development and national policies) that could affect CRF. The elasticity of the impact of PCDI on CRF of urban girls (aged 13–22 years) is −0.022 and not significant (*p* > 0.1), indicating that the relationship between PCDI and CRF of urban girls (aged 13–22 years) is not linear.

#### 3.2.2. Quadratic Model

To further explore trends of influence of economy on students’ CRF levels, the square term of LPCDI was added to test the marginal effect [26]. Columns (quadratic) in Table 4 and Table 5 show the results of the nonlinear relationship test between economic growth and CRF of children and adolescents, as estimates of Model (2). The results show that β1 > 0 and β2 < 0 and both are statistically significant, indicating that PCDI has an inverted U-shaped relationship with the mean running test time, that is, it has a U-shaped relationship with cardiopulmonary endurance levels, especially for children, boys (β1 = 0.328 (*p* < 0.01); β2 = −0.018 (*p* < 0.01)) and girls (β1 = 0.347 (*p* < 0.01); β2 = −0.019 (*p* < 0.01)). It appears that as incomes increased, the CRF of urban male and female students gradually decreased to the lowest point, after which it showed an upward trend. From a horizontal perspective, it can be inferred that for less-developed provinces, increases in incomes cause a decrease in CRF levels; subsequently increasing health risks. In contrast, for highly developed provinces, as incomes increase, CRF increases.

## 4. Discussion

This study shows a pattern of CRF–Kuznets curve, a U-shaped relationship between economy and CRF of children and adolescents based on provincial data from 1985–2014. The analysis reveals that the negative impact of economic development on cardiorespiratory fitness of children and adolescents gradually decreased over the years. Similarly, another study found a non-linear relationship between a country’s income per capita and weight-related health status (Obesity Kuznets curve) for men and women according to the country-level panel data of 130 countries from 1975 to 2010 [28]. The Obesity Kuznets curve has also been reported for state-level panel data from 1991–2010 [29]. However, few studies have explored the relationship between macroeconomics and cardiopulmonary fitness. 

Our results reveal that as an economy grows, the cardiorespiratory fitness level of students aged 7–22 years gradually decreases; this decline tends to slow down and stagnate at some point. Indeed, from 1984 to 2014, the CRF of Chinese urban students showed a U-shaped trajectory. From 1984 to 2005, national survey reports on students’ physical health showed that the CRF of children and adolescents continuously decreased [22,30,31], as shown in Table 3. However, in 2010, a national survey on student physical fitness showed that the continuous decline in CRF of primary and middle school students had been contained [32]. This is because running time of the 50 m × 8 was shorter by 0.05 s on average for girls and unchanged for boys aged 7–12 years, compared to 2005. The running time of boys and girls aged 13–15 years in junior middle schools decreased by 3.03 s and 3.58 s on average, respectively. The average performance of boys and girls aged 16–18 years decreased by 0.48 s and 0.46 s, respectively, compared with 2005 [32], as shown in Table 2 and Table 3. A national student physical fitness survey conducted in 2014 showed that the cardiorespiratory fitness of primary and middle school students remained stable [33].

The decrease in CRF of children and adolescents from 1984 to 2005 may be explained by the following factors. First, over-nutrition affects cardiorespiratory endurance. A study found that the CRF of overweight and obese students was significantly lower than that of students with normal weight [27]. Moreover, a higher BMI is associated with decreased cardiorespiratory endurance levels [27]. This indicates that being overweight and obese negatively affects CRF of students. It is conceivable that the decline in Chinese students’ CRF over the past 30 years is due to an increase in overweight and obesity rates. In addition, students’ sedentary lifestyles may affect their cardiopulmonary fitness [34,35]. Owing to rapid developments in science and technology, China has entered an era of automation—private cars, buses, subways, and other means of transportation have decreased walking. The diversification and modernization of modes of transport have decreased participation in physical activity among teenagers, leading to underutilization of human energy [36]. In addition, the exam-oriented education system and high academic demands have reduced the amount of leisure time for students. In addition, several students tend to choose electronic games over physical activity for leisure [37]. These factors are likely to affect CRF of students.

However, data has shown that from 2005 to 2014, the deterioration in cardiorespiratory endurance levels of students has been contained, especially for middle and high school students. This can be explained by the following reasons: First, several community sports facilities, national fitness centers, fitness paths, fitness squares, and parks have been built, providing teenagers with better platforms to engage in physical activity [38]. Second, family incomes and parents’ education levels have also increased. Some studies have shown that the education level, health awareness, and financial support of parents have a positive impact on children’s and adolescents’ participation in physical activity or physical education [39,40,41]. This is because as the social economy develops and living standards improve, people tend to pay more attention to physical and mental health, especially with respect to children.

This study examined the relationship between economic development and CRF of children and adolescents. This analysis is based on one million data collected from surveys conducted in five-year interval periods between 1985 and 2014 covering 27 provinces in China. We show that the relationship between economic development and CRF of children and adolescents is U-shaped. For low-income provinces, increases in income causes a decrease in the CRF-related health status. In contrast, for high-income provinces, as incomes increase, the CRF-related health status improves. Our findings support the possibility that youths living in middle-income provinces may be at risk of poor health, which calls for health policies targeting prevention and intervention in China and other developing countries.

The findings of this study may be affected by endogeneity. Another limitation is that cardiorespiratory fitness was measured by a long-distance running test, which may be affected by several factors, such as environmental conditions and running surfaces. However, in order to collect time series data for further studies, physical fitness in China was measured over a long period. Previous studies suggest that running tests are valid and reliable. The main strength of this study is the large sample size used, based on five-year assessments of CRF for children and adolescents aged 7–22 years over a 30-year period. This study also provides evidence of the impact of social environmental factors (such as economy and policy) on cardiorespiratory fitness of children and adolescent based on panel data. 

Future multi-level studies based on the theory of social ecological model should be conducted. Other studies should explore the impact of national economy on the health of rural Chinese students, to compare cardiorespiratory fitness between urban and rural areas. In order to formulate more effective and scientific intervention measures, studies should be conducted from a socio-ecological microsystem perspective, such as correlation and collaboration among schools, communities, and families.

## 5. Conclusions

The main finding of this study is that a U-shaped relationship exists between China’s economic development and the cardiorespiratory fitness of children and adolescents. The negative effect of economic development on cardiorespiratory fitness of urban students is seen to decrease and eventually reach stagnation, especially in highly developed provinces. This analysis provides evidence that students living in middle-income provinces may be at risk of developing health problems, calling for effective health policies targeting prevention and intervention. 

## Figures and Tables

**Table 1 ijerph-16-03772-t001:** Sample Sizes at Each Examination Period in CNSSCHA (Aged 7–22 years), 1985–2014.

Age	Male							Female						
	1985	1991	1995	2000	2005	2010	2014	1985	1991	1995	2000	2005	2010	2014
7	8560	2933	4401	3051	4935	4486	4464	8559	2925	4400	4512	4860	4481	4480
8	8561	2999	4398	4446	4934	4485	4489	8561	2977	4406	4533	4862	4469	4487
9	8557	2959	4397	4546	4945	4481	4478	8561	2951	4402	4494	4892	4490	4490
10	8557	2932	4390	4571	4916	4490	4485	8559	2942	4402	4657	4932	4487	4488
11	8561	2965	4396	4587	5051	4500	4487	8559	2935	4413	4498	4894	4496	4474
12	8558	2957	4398	4524	4917	4483	4479	8557	2875	4397	4518	4785	4484	4477
13	8558	2906	4394	4523	4914	4487	4487	8558	2933	4395	4530	4925	4487	4495
14	8560	2948	4403	4546	4852	4489	4487	8561	2931	4383	4513	4859	4494	4485
15	8556	2952	4403	4561	4979	4489	4482	8556	2924	4394	4516	4935	4482	4489
16	8559	2936	4343	4551	4892	4476	4481	8557	2911	4396	4519	4916	4454	4493
17	8532	2936	4395	4507	4919	4488	4494	8537	2910	4403	4524	4857	4485	4490
18	8324	2968	4335	4549	4979	4476	4289	8159	2933	4365	4632	5047	4435	4289
19	5516	2937	2850	3302	3777	2971	2973	5551	2988	2863	3449	3938	2996	2997
20	5593	2953	2876	3181	3741	2975	2970	5580	2968	2880	3306	3905	2982	2988
21	5592	2770	2875	3187	3744	2972	2989	5462	2751	2879	2958	3752	2986	2987
22	4332	2955	2829	2718	3327	2915	2865	3335	2685	2737	2555	3390	2917	2881
Total	123,476	47,006	64,083	65,350	73,822	65,663	65,399	122,212	46,539	64,115	66,714	73,749	65,625	65,490

**Table 2 ijerph-16-03772-t002:** Summary Statistics of all Variables.

Variable	Observations	Mean	S.D.	Min	Max
50 m × 8 (Both)	2268	123.63	10.28	95.66	184.82
50 m × 8 (Male)	1134	121.23	10.26	95.66	184.82
50 m × 8 (Female)	1134	126.03	9.72	102.73	184.79
1000 m	1890	258.09	19.72	216.73	348.15
800 m	1890	252.27	16.31	202.65	320.93
PCDI	6048	9696.99	9769.57	510.14	47,758.17
URBAN	6048	40.82	18.13	11.48	89.6
CPI	6048	105.72	5.89	97.9	121.4
L50 × 8 (Both)	2268	4.81	0.08	4.56	5.22
L50 × 8 (Male)	1134	4.79	0.08	4.56	5.22
L50 × 8 (Female)	1134	4.83	0.08	4.63	5.22
L1000 (Male)	1890	5.55	0.08	5.38	5.85
L800 (Female)	1890	5.53	0.06	5.31	5.77
LPCDI	6048	8.55	1.24	6.23	10.77
LURBAN	6048	3.6	0.48	2.44	4.5
LCPI	6048	4.66	0.05	4.58	4.8

PCDI, real per capita disposable income; URBAN, urbanization rate; CPI, consumer price index; L50 × 8, log of 50 m × 8; L1000, log of 1000 m; L800, log of 800 m; LPCDI, log of real PCDI; LURBAN, log of URBAN; LCPI, log of CPI.

**Table 3 ijerph-16-03772-t003:** Running Time for Children and Adolescents (s), 1985–2014

Sex (years)	7	12	13	16	19
Male					
1985	125.48	104.58	265.57	240.90	232.91
1991	128.40	106.18	268.74	246.64	234.83
1995	128.11	107.35	272.41	246.33	233.93
2000	135.77	115.70	283.42	253.73	245.26
2005	137.88	116.18	299.52	266.31	258.41
2010	136.63	114.69	296.92	265.66	259.02
2014	136.60	115.71	302.67	267.51	264.18
Female					
1985	131.44	112.30	238.84	241.15	234.47
1991	133.40	113.13	241.74	243.64	235.99
1995	132.63	114.14	243.50	243.50	233.29
2000	140.08	123.91	260.00	253.56	247.38
2005	141.03	125.11	273.24	264.36	254.24
2010	139.94	121.66	270.98	263.24	257.57
2014	140.27	121.68	269.54	263.74	255.84

50 m × 8 for boys and girls (7–12 years), 1000 m for boys (13–22 years), and 800 m for girls (13–22 years).

**Table 4 ijerph-16-03772-t004:** Econometric Estimates of the Relationship Between PCDI and CRF of Children.

Variable	L50 × 8		L50 × 8	
	Linear (Male)	Quadratic (Male)	Linear (Female)	Quadratic (Female)
LPCDI	−0.006	0.328 ***	−0.001	0.347 ***
	(−0.35)	(5.45)	(−0.07)	(6.14)
LPCDI^2^		−0.018 ***		−0.019 ***
		(−5.71)		(−6.38)
LURBAN	−0.020 ***	−0.013 **	−0.021 ***	−0.014 **
	(−2.77)	(−2.03)	(−2.93)	(−2.13)
LCPI	0.204 **	0.025	0.211 **	0.024
	(2.50)	(0.35)	(2.41)	(0.30)
_cons	3.872 ***	3.302 ***	3.866 ***	3.270 ***
	(8.79)	(7.11)	(8.49)	(6.81)
R^2^	0.547	0.565	0.466	0.491
F	159.471	147.363	106.812	100.308
*P*	0.000	0.000	0.000	0.000
*N*	1134	1134	1134	1134

L50 × 8, log of running time for 50 m × 8; LPCDI, log of real per capita disposable income; LPCDI^2^, the squared term of LPCDI. LURBAN, log of urbanization rate; LCPI, log of consumer price index; The White (1980) robust regression. t statistics in parentheses. *, ** and *** denote statistical significance at the 10%, 5%, and 1% levels, respectively.

**Table 5 ijerph-16-03772-t005:** Econometric Estimates of the Relationship Between PCDI and CRF of adolescents.

Variable	L1000		L800	
	Linear (Male)	Quadratic (Male)	Linear (Female)	Quadratic (Female)
LPCDI	−0.027 **	0.093 **	−0.022	0.116 **
	(−2.25)	(2.05)	(−1.53)	(2.52)
LPCDI^2^		−0.007 ***		−0.008 ***
		(−2.72)		(−3.07)
LURBAN	−0.001	0.001	0.000	0.003
	(−0.23)	(0.20)	(0.05)	(0.45)
LCPI	0.010	−0.054	0.124	0.049
	(0.15)	(−0.82)	(1.50)	(0.59)
_cons	5.628 ***	5.423 ***	5.045 ***	4.808 ***
	(16.17)	(14.44)	(11.86)	(10.94)
R^2^	0.615	0.617	0.514	0.518
F	226.681	225.073	138.782	131.749
*P*	0.000	0.000	0.000	0.000
*N*	1890	1890	1890	1890

L1000, log of running time for 1000 m (boys); L800, log of running time for 800 m (girls); LPCDI, log of real per capita disposable income; LPCDI^2^, the squared term of LPCDI. LURBAN, log of urbanization rate; LCPI, log of consumer price index; The White (1980) robust regression. t statistics in parentheses. *, **and *** denote statistical significance at the 10%, 5%, and 1% levels, respectively.

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
