# Peer review of "Economic Growth and Cardiorespiratory Fitness of Children and Adolescents in Urban Areas: A Panel Data Analysis of 27 Provinces in China, 1985–2014"

_ijerph, 2019, doi:10.3390/ijerph16193772_

Round 1
Reviewer 1 Report
Thank you very much for submitting this manuscript, which I read with interest. The document is otherwise well written, providing a clear introduction, as well a sound explanation for the included statements.
The abstract clearly defines what the purpose of the study is, however the author(s) should mention the following factors:
(i) More important contributions and implications of this study.
(ii) The original value for this field of study.
Introduction:
(i) The introduction presents the motivations/reasons which lead to the study but does not highlight the innovative aspect of this research.
(ii) The research goals are poorly defined.
Discussion and conclusions:
The discussion and conclusions are well but the implications and contributions are mixed, lacking a better specific argumentation, as well as the innovative aspect of this study
References:
Bibliography seems well but more up-to-dated references are recommended
Author Response
Response to Reviewer 1 Comments
Comment:
Thank you very much for submitting this manuscript, which I read with interest. The document is otherwise well written, providing a clear introduction, as well a sound explanation for the included statements.
Response:
Thank you for providing the detailed and constructive suggestions for our manuscript. We have carefully reviewed, addressed each comment below and made corresponding changes in the manuscript (highlighted in yellow).
Point 1: The abstract clearly defines what the purpose of the study is, however the author(s) should mention the following factors:
(i) More important contributions and implications of this study.
(ii) The original value for this field of study.
Response 1:
Thank you for providing constructive suggestions .We apologize for unclear description in the manuscript, Revision was made in the manuscript. (page 1,line 25-29)
Point 2: Introduction:
(i) The introduction presents the motivations/reasons which lead to the study but does not highlight the innovative aspect of this research.
(ii) The research goals are poorly defined
Response 2:
(i) Thank you for the constructive suggestions! Revision was made in the manuscript. (page 2,line 21-25)
(ii) Thank you for the constructive suggestions! Revision was made in the manuscript. (page 2,line 18-21)
Point 3: Discussion and conclusions:
The discussion and conclusions are well but the implications and contributions are mixed, lacking a better specific argumentation, as well as the innovative aspect of this study
Response3:
We apologize for that implications and contributions were mixed, Revision for implications and contributions was made in the manuscript. (page 8,line 18-26)
Revision for specific argumentation and innovation was made in the manuscript. (page 7,line 5-13)
Point 4: References: Bibliography seems well but more up-to-dated references are recommended
Response4: Thank you for the constructive suggestions! Revision was made in the manuscript. (page 9,line 8; page 9,line39 and 42).
Reviewer 2 Report
Major comments:
This study demonstrated a U-shaped relationship between Chinese economic development and the cardiorespiratory fitness of children and adolescents. However, the authors should debate why they did not utilize multi-level models to deal with the ecological fallacy.
Minor points:
Line 36, page 3: where is Fig. 2? Table 5: why are there redundant columns of Linear(male)/ Quadratic(female)? Table 6: Under the column L1000, Quadratic(female) should be switched to Quadratic(male); under the column L800, Linear(male) should be switched to Linear(female).Author Response
Response to Reviewer 2 Comments
Major comments:
This study demonstrated a U-shaped relationship between Chinese economic development and the cardiorespiratory fitness of children and adolescents. However, the authors should debate why they did not utilize multi-level models to deal with the ecological fallacy.
Response:
We apologize for why select the fixed effect model regression as estimation method was not explained. The fixed-effects model was chosen because time-invariant variables were not included in this study. And the advantage of the fixed effect is that the relationship between the error term and the explanatory variable need not be assumed, which Multi-level model do not have. We made the revision in the manuscript. (page4, line 1-3)
Minor comments:
Line 36, page 3: where is Fig. 2? Table 5: why are there redundant columns of Linear(male)/ Quadratic(female)? Table 6: Under the column L1000, Quadratic(female) should be switched to Quadratic(male); under the column L800, Linear(male) should be switched to Linear(female).
Response:
Thank you for providing detailed suggestions .We apologize for the mistakes in the manuscript, Revision was made in the table 4and 5.
Reviewer 3 Report
The study aimed to assess the relationships between China’s macro-economy developments and cardiorespiratory fitness of children and adolescents over the past 30 years on 1 million samples. The results show that economic growth may have "U" shape relationships with CRF of children and adolescents (for both boys and girls). The paper has certain implication. However several changes are needed:
A short background statement should be added in the abstract.
The introduction is broad and long, it should be shortened and focused.
The number and date of the ethical committee approval should be mentioned in the method section.
The statistical analysis section is unclear and should be improved and better described.
The discussion section should be re-written and as follows:
The finding(s) of the current study and comparison with previous published (similar) studies The implication of the findings The strengths and limitations of the study The new direction of the future research
The number of tables should be reduced.
The English need to be edited and revised.
Author Response
Response to Reviewer 3 Comments
Comment:
The study aimed to assess the relationships between China’s macro-economy developments and cardiorespiratory fitness of children and adolescents over the past 30 years on 1 million samples. The results show that economic growth may have "U" shape relationships with CRF of children and adolescents (for both boys and girls). The paper has certain implication. However several changes are needed.
Response:
Thank you for providing the detailed and constructive suggestions for our manuscript. We have carefully reviewed, addressed each comment below and made corresponding changes in the manuscript (highlighted in yellow).
Point 1: A short background statement should be added in the abstract.
Response 1: Thank you for providing constructive suggestions. A short background statement has been added in the abstract. Revision was made in the manuscript. (page 1,line 10-13)
Point 2: The introduction is broad and long, it should be shortened and focused.
Response 2:
Thank you for the constructive suggestions! The introduction has been shortened and focused. Revision was made in the manuscript. (page 1,line 33-page 2,line 9)
Point 3: The number and date of the ethical committee approval should be mentioned in the method section.
Response3:
We apologize for unclear description in the manuscript. The research protocol was reviewed and approved by the Ethics Committee of College of Education at Zhejiang University (March 28th, 2019).The number is not available. Revision was made in the manuscript. (page 2,line 44-45)
Point 4: The statistical analysis section is unclear and should be improved and better described.
Response4:
Thank you for the constructive suggestions! Revision was made in the manuscript. (page 4,line 1-36).
Point 5: The discussion section should be re-written and as follows:
The finding(s) of the current study and comparison with previous published (similar) studies The implication of the findings The strengths and limitations of the study The new direction of the future research
Response5:
Thank you for the constructive suggestions! Revision was made in the discussion. (page7, line 5-page8,line 41).
Point 6: The number of tables should be reduced.
Response6:
Thank you for the constructive suggestions! We combined table 3 and table 4 in the first draft into a single table named table 3. Revision was made in the manuscript. (page 6,line 3).
Point 7: The English need to be edited and revised.
Response7:
We apologized for the English.
The manuscript has been revised by a professional English Editing Company.
We hope you find the changes satisfactory.
Round 2
Reviewer 1 Report
The manuscript could get more relevant including your additional work. I wish the present study would influence to trends in cardiorespiratory fitness (CRF) of Chinese children.
Reviewer 2 Report
Much better.
Reviewer 3 Report
I would like to thank the authors for being highly responsive. I think that now the manuscript is suitable for publication in the current form.